# Research on Dynamic and Mechanical Properties of Magnetoactive Elastomers with High Permeability Magnetic Filling Agent at Complex Magneto-Temperature Exposure

**DOI:** 10.3390/ma14092376

**Published:** 2021-05-03

**Authors:** Maria Vasilyeva, Dmitriy Nagornov, Georgiy Orlov

**Affiliations:** 1Department of Transport and Technological Processes and Production, Saint-Petersburg Mining University, 199106 Saint-Petersburg, Russia; 2Department of Geoecology, Saint-Petersburg Mining University, 199106 Saint-Petersburg, Russia; Nagornov_DO@pers.spmi.ru; 3Department of Automation of Technological Processes and Production, Saint-Petersburg Mining University, 199106 Saint-Petersburg, Russia; Orlov_GA@pers.spmi.ru

**Keywords:** composite polymers, magnetoactive elastomers, dynamic and mechanical properties, concentration, anisotropy, fineness

## Abstract

We consider magnetically active elastomer as a potentially applicable material for manufacturing a working channel of a magnetic pump unit. During the study, the samples were exposed to a magnetic field, a temperature field, and their combination to assess the change in the elastic-strength properties of the final material. For the preparation of samples, high permeability magnetic fillers of various sizes were used in the concentration range of 50–70%. Samples were made with an isotropic and an anisotropic structure. Studies have shown that when using a filler with a relatively coarse fraction, the material has more stable dynamic and mechanical characteristics: the tensile strength of the sample increases by an average of 38%. With the combined effect of magnetic and temperature fields on the material, its elasticity and strength increase by an average of 30% in comparison with the material without external influence. Based on the results obtained, the composition and structural organization of the material, which has the best complex of elastic strength characteristics, has been substantiated. For the manufacture of a pumping unit tube, it is preferable to use an isotropic magnetoactive elastomer with a coarser filler content of about 60%.

## 1. Introduction

Magnetoactive elastomers (MAEs) are a new class of materials. Under the influence of an external magnetic field, they are able to change their mechanical properties and linear dimensions. This material contains a strong magnetic dipole interaction of the filler particles. It depends on the magnitude of the saturation magnetization. This leads to the use of materials with a high content of iron or cobalt as a filler [1,2,3,4]. Due to the magnetorheological properties of this material, in recent years, its applications for aerospace, automotive, construction, and medicine have been widely studied [4,5,6,7,8,9].

One of the upcoming trends of application of magnetoactive elastomers is the nature-like magnetic pump being developed at the St. Petersburg Mining University (St. Petersburg, Russia). The working channel of this pump is made of MAE. The area of its free cross-section is based on the properties of the transported substance (density, viscosity, etc.). The pump unit activator generates electromagnetic pulses according to the algorithm set. The MAE working channel is deformed under the influence of an external magnetic field. The process of successive contraction of the working channel sections generates a moving deformation wave and makes it possible to implement the principle of peristaltic transportation in the required direction [10,11]. Such pumping units mostly do not have the inherent disadvantages of traditional volumetric units, to be more specific, its performance depends on the features of the electromagnetic field source (velocity of the local deformation wave, power), and eliminates pressure losses caused by the pumped medium properties [12]. Due to additional use of the working channel inner surface relief the flow can be structured and delamination of inhomogeneous media can be prevented.

During the operation of the pump unit, the tube must maintain integrity. To substantiate the most suitable set of elastic-strength characteristics of the material, the following was comprehensively studied: the effect of the MAE composition components, the effect of the percentage of the filler, the effect of the ordering of its structure, the effect of external magnetic and temperature fields.

## 2. Materials and Methods

### 2.1. Elastomer Material

For the manufacture of MAEs, polymers based on vinyl or rubber are used as a matrix [13,14,15], and microscopic magnetoactive particles are used as a filler [16,17].

For the purposes of this research, a two-component silicone compound—UNISIL 9115—has been used, the compound had the following specifications: viscosity—15,000 mPa·s, Shore hardness—15 A, elasticity—500%, tensile strength—3.5 MPa, split sample tear resistance—12 kN/m, linear shrinkage—1%.

The two following types of high permeability magnetic materials were used as filling agents:

—atomized powdered iron (complies with GOST 9849-86): bulk density–2.8 g/cm^3^, grain-size composition basic fraction–150 µm. The relative magnetic permeability—µ/µ_0_ = 5.0 × 10^3^.

—carbonyl iron—R-10 (complies with GOST 13610-79): bulk density 2.0 g/cm^3^, grain-size composition basic fraction—5 µm. The relative magnetic permeability—15 μ/μ_0_.

The compound was mixed by ferromagnetic particles to yield MAEs with the filling content of each type at 50%, 60%, and 70%, respectively. Furthermore, the catalyst was added in a 100:3 ratio and then the mixture was degassed. Then, depending on the planned type of experiment, the mixture was placed in special molds until complete polymerization. They were made individually for each experiment using 3D printing.

The choice of the range of filler content in the composition of MAE samples was based on the results of studies of the dependence of the relative (dielectric) permittivity and magnetic pressure (attraction of the tube walls) distribution (Figure 1a,b) [18,19,20,21].

Samples with concentrations above 70% cause difficulty to ensure homogeneous distribution of filler particles in the polymerization step because of the rapid sedimentation of the filler particles. Also, in such samples, the tensile and compressive strength of elastomers decreases due to a change in the bond strength at the polymer–filler interface with an increase in the particle size [22,23]. Above 80%, the structure of the elastomer corresponds to an inverse dispersion.

The samples were divided into two groups, equal in the number of samples. One group in the process of polymerization was exposed to an external magnetic field of 1.2 T for 12 h. This resulted in samples with distinct structure anisotropy along the magnetic field lines direction (Figure 2c,d). The second group of samples was not subjected to further exposure in course of curing process and exhibited an isotropic structure [24,25,26,27].

In order to assess the mixing quality and the degree of anisotropy intensity, the research was carried out on a JSM-6460 scanning electron microscope (JEOL) in SEI and BEI mode (accelerating voltage—20 kV, current n × 10^−10^ A) [28]. The COMPO mode was the most effective signal for the task. The samples were preliminarily fastened with carbon glue and dried for several hours under an air stream due to the material properties. The samples were pre-coated with gold for stable conductivity.

### 2.2. Magnetic Field Source

In order to conduct experimental studies, samples of three different shapes were made for each set of composition and components structure. The casting molds were produced with 3D printing.

When determining the material for the manufacture of a pumping unit tube, it is necessary to consider many factors: the provided pump head, the permissible pressure on the tube wall, as well as the permissible gaps when the walls are closed [29].

A neodymium magnet was used as the magnetic field source. The loss of magnetic properties of such magnets is not more than 2–3% over 10 years of operation (if the allowable temperature conditions are observed). The magnet is made in the shape of a 10 × 10 × 40 mm bar (Figure 3). It is protected by a nickel coating to protect against the adverse effects of corrosion and rust. The sharp edges of the magnetic plate have been beveled. The magnetic field direction is axial. The magnet properties are shown in Table 1.

### 2.3. Study of Dynamic and Mechanical Properties of MAE

The sample shape and type, along with its production method depends on the material composition, the choice was made in accordance with GOST 270-75 Rubber. Method of the determination elastic and tensile stress-strain properties. For this study, the samples were formed in double-sided blades with the appropriate set of dimensions (Figure 3, Table 2). The MAE plates of (2.0 ± 0.2) mm thickness with different concentrations and material organization structure were prepared. After a period of polymerization, the samples were cut out of them with specimen die.

Except for thickness, the sample dimensions were based on cutters’ dimensions and left unchecked after cutting. To ensure equal installation of the samples, installation marks were put in the grips, the distance between the marks was (40 ± 1) mm.

The research of the MAE stress-related properties under tension was estimated in terms of tensile strength values, elongation at rupture, and stress at a given elongation. The experimental procedures comprised three stages (Figure 4):

A—tests at a temperature of (23 ± 2) °C and active grip motion velocity—500 mm/min;

B—tests at a temperature of (23 ± 2) °C, application of an external magnetic field with the strength of 1.2 T and active grip motion velocity—500 mm/min;

C—tests at a temperature of (70 ± 2) °C, application of an external magnetic field with the strength of 1.2 T and active grip motion velocity—500 mm/min.

Due to the elastomer’s viscoelastic material behavior, the surface properties depend on loading speed and temperature. Different peeling speeds result in different adherence strength of an interface between the gripper and the substrate [30]. The studies were carried out at the same speed to exclude the influence of this effect.

Studies at elevated temperature were conducted in a special thermal chamber. It ensured the required temperature in the operating volume (limited by the machine traverse grips) with a permissible error of ± 2.0 °C. The temperature was monitored at three points on the elastomer neck surface by the Fluke Ti450 thermal imager.

## 3. Results and Discussion

### 3.1. Assessment of the Material Adsorption Capacity

Studies of the impact of the external magnetic field strength during the MAE polymerization at variable concentrations of the filling agent, the prepared samples with the filler content of 50%, 60%, and 70% were researched by SEM analysis.

The MAE containing carbonyl iron particles has a spherical filling agent with a size of up to 5 μm. In samples exposed to a magnetic field, the distribution of these particles in the matrix is different from a sample of the same concentration, polymerized without such exposure (Figure 5).

In sample (b), the particles are aligned in elongated agglomerates parallel to the lines of action of the magnetic field; the distance between these lines is similar. The sample (d) changes its structure in a similar way. However, it is difficult to evaluate the anisotropic disposition of the particles in this sample in areas with increased agglomeration of iron particles that can range in size from n × 10 µm to 1 mm. In the sample (e) without magnetic impact, areas of iron particles agglomeration, 100–300 µm in size, and formation of ‘sphere-in-sphere’ structures, sized about 10 µm, can also be observed. After magnetic impact, the anisotropic structure with ‘sphere-in-sphere’ structures is denser; however, it acts differently in the agglomeration areas (Figure 6).

The frequency of such agglomerations significantly increases with the increase in concentration of the filling agent, it also increases their linear size. A more detailed consideration of such structures reveals an arbitrarily shaped core consisting of large particles of the filling agent [31]. Furthermore, the smaller particles are organized in an orderly manner around the larger ones forming a spatial layered structure. This effect can be explained by the higher value of the effective magnetic field that appears as a combination of the external field and the self-induced field of large particles of the filling agent.

An increase in agglomeration ability along with a decrease in particle size and then an increase in agglomerate size is the result of the adsorption capacity [32]. The greater the tendency to agglomeration, the lower the value of the limiting filling of the volume φ_max_, Also, with an increase in the adsorption capacity, the properties of the composite material change more intensively.

The formation of an anisotropic structure is pronounced in the MAE samples prepared with the use of atomized powdered iron as a filling agent (Figure 7).

While the chemical composition of R-10 and atomized powdered iron fillers is identical, and despite the MAE preparation method, the size and weight of the solid fraction particles determine the differences in the structural organization of the final material.

The analysis of the distribution of the filler in the matrix material was carried out using the ToupView software. It allowed the creation of a three-dimensional representation of the intensity of the image. Redistribution along the selected plane made it possible to obtain an image reflecting the levels of distribution of the filler in the sample (Figure 8). The images clearly show the formation of ‘sphere in sphere’ structures. They are arranged in columnar structures, merging into a common agglomerate, growing in the direction of the source of the magnetic field.

This kind of structural organization certainly affects the material mechanical properties resulting in a loss of stability when the content of the magnetic filling agent reaches 70% and higher (Figure 9).

### 3.2. Assessment of the Dynamic and Mechanical Properties of the Material

When comparing fillers, it should be noted that the reduction in the filler grains size leads to an increase in their total specific surface area, increase in the number of particles at the same volumetric content, and, consequently, to the decrease in the distance between the filler particles in the portable optically pumped magnetometer and increase in their ability to form agglomerates.

The follow-on tests made it possible to evaluate the effect of the filler concentration and its structure on the main indicators of the elastic-strength properties of elastomers. Considering the principle of operation of the pumping unit, the main ones were modulus of elasticity, viscosity, strength of MAE.

By calculation, the elastic modulus of a polymer mixture can be predicted only roughly. The best fit of the equation to the experimental data is observed in the interval of the filler concentration up to 20% and above 80%. For the middle concentration range, where phase reversal occurs, the calculations are of little use (Figure 9) [33].

Solid non-deformable filler particles insertion reduces the deformability of MAE with an increase in the filler content, as well as an increase in the particle size (Figure 10).

The development curves of the elastomers’ elastic modulus indicate that MAE with atomized powdered iron filling has on average 50% higher initial elasticity compared to MAE with R-10 filler. Pairwise comparison of the samples showed that the plasticity of an anisotropic sample almost does not differ from an isotropic one.

Evaluation of the change in the toughness of the material when exposed to a magnetic field revealed a stable increase of this parameter. Under the combined effect of the magnetic and temperature field, the toughness of the material decreases slightly due to softening of the matrix material.

The value of the phase contact area in a composite elastomer is determined by the filler’s specific surface area [34]. When the particle size is less than 10 μm, a sharp decrease in the distance between particles in the concentration range above 60% is observed.

When the material is exposed to magnetic and temperature fields, the greatest amplifying effect is recorded when the interfacial layers in the material are in contact. This condition is met when a filling agent with a larger fraction (atomized powdered iron) is applied.

Comparing the same length stretching of pure silicone and silicone-based MAE samples, it must be noted that the matrix by itself (with no solid filler involved) should provide the required deformation. Therefore, the matrix deforms to a greater extent than the pure polymer when the filler content increases (*φ*)
(1)$$\varepsilon_{sil}=\frac{\varepsilon_{MAE}}{1-\varphi }$$
where $\varepsilon_{sil}$—silicon matrix elongation.

Accordingly, at 50% filler particles in the MAE composition, the matrix deforms by 4 times when such a sample is elongated by 2 times, and by 6 times at 70% [35]. As the results of experiments have shown, this inevitably leads to the MAE destruction (Figure 11).

The strength of the composite elastomer correlates worse with the change in the modulus of elasticity with increasing filler content. This can be explained by the fact that the strength derives from the conditions of microdefect propagation at fracture. Whereas the modulus is a property peculiar to the undamaged structure [36]. The development of shrinkage internal stresses or stress buildup on the filler particles in the deformation process greatly affects the strength and hardly affects the modulus [37].

Deformation of the matrix polymer in combination with periodic loading of the material, overvoltage at the matrix–filler interface will inevitably lead to the appearance of microdefects in the MAE. As the relative degree of filling increases to 60–70%, the overstressed areas around the particles begin to overlap facilitating porosity growth, and the material strength decreases. This effect is attributed to a decrease in the bond strength at the silicone–filler interface. The decrease in the strength of the material with the P10 filler by an average of 38% is explained by the smaller distance between the particles.

This phenomenon also correlates with the Blanchard–Parkinson’s theory that explains the polymer composites reinforcement through the strengthening effect caused by the formation of strong bonds ‘silicone-filling agent’ [38]. Since the number of these bonds is determined by size and nature of the filler surface, the filler particles with a relatively larger contact surface area ensure good reinforcing properties.

An assessment of the change in the strength of MAE under the influence of magnetic and temperature fields revealed that the ultimate strength of anisotropic samples is noticeably reduced in comparison with an isotropic material. The filler particles line up in chains, thus creating zones with an uneven thickness of the interfacial layer.

## 4. Conclusions

A comprehensive study made it possible to determine the parameters of an elastomer suitable for the manufacture of a tube of a peristaltic pump unit. Based on the results of evaluating the elastic-strength properties of magnetoactive elastomers based on soft magnetic fillers, the following conclusions were drawn:

1. An increase in the filler concentration in the MAE composition promotes a deformability reduction. This effect is amplified when exposed to an external magnetic field, and slightly attenuates when exposed to both magnetic and temperature fields at the same time. This attenuation effect can be explained by the faster migration of the filling agent particles, hooks, and nodes of the material structure exposed to the elevated temperature.

The filling agent content above 70% leads to a decrease in the stability of the final material and is not recommended for use.

2. The anisotropic structure of the MAE leads to an enhancement of the samples’ elastic properties amplified when exposed to an external magnetic field. It also contributes to the formation of agglomerates in the final material, which leads to a drop in strength properties. This trend exponentially increases as the filling agent content of the matrix increases and the grain size of the material decreases.

3. The magnetoactive elastomer prepared with a filler that has a relatively larger fraction exhibits more stable dynamic and mechanical properties—the ultimate strength of the sample rises by 38% on average, further growing with such raise; the elastic modulus is two-fold higher (Figure 2 and Figure 3).

4. When MAE samples are exposed to an external magnetic field, regardless of the filler size, the samples elasticity and toughness increase. This effect is particularly significant at lower concentrations. With the combined effect of magnetic and temperature fields, the elasticity and toughness of materials increases by an average of 30% in comparison with the material without external influence (Figure 2 and Figure 3).

5. The conducted studies indicate that MAEs with the content of filling agent about 60%, with relatively larger filler fraction and isotropic structure of the final polymerized material are recommended as the most suitable material for producing pumping equipment parts (Figure 1).

## Figures and Tables

**Figure 1 materials-14-02376-f001:**
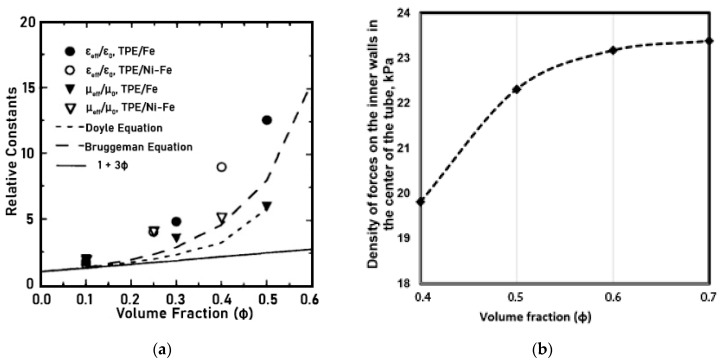
Properties of the MAEs as a function of the filler content: (**a**) the dependence of the dielectric constant; (**b**) the dependence of the distribution of pressure (magnetic attraction) in the tube.

**Figure 2 materials-14-02376-f002:**
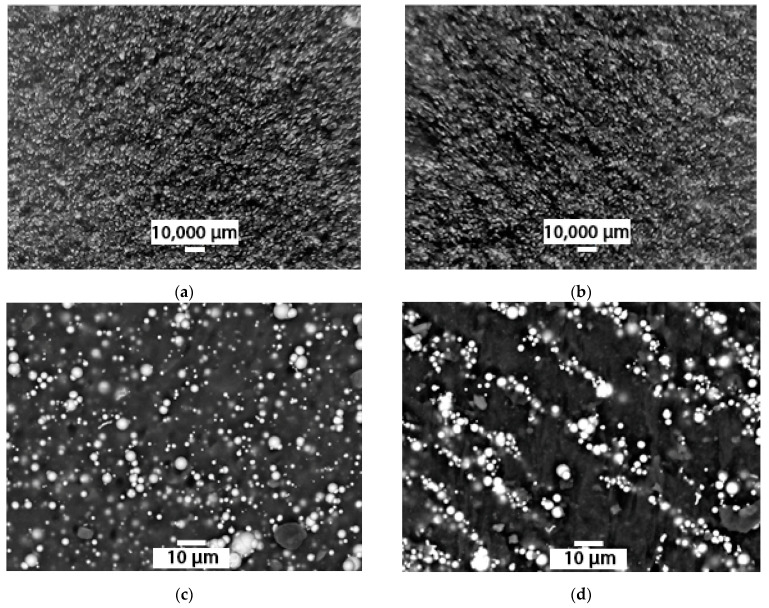
Filling agent particle distribution in the polymer matrix MAE*: Atomized powdered iron filler: (**a**) isotropic sample, (**b**) anisotropic sample; R-10 carbonyl iron filling agent: (**c**)—isotropic sample, (**d**) anisotropic sample. * Due to the large difference in the filling agent particle sizes, different image scales have been chosen by way of illustration.

**Figure 3 materials-14-02376-f003:**
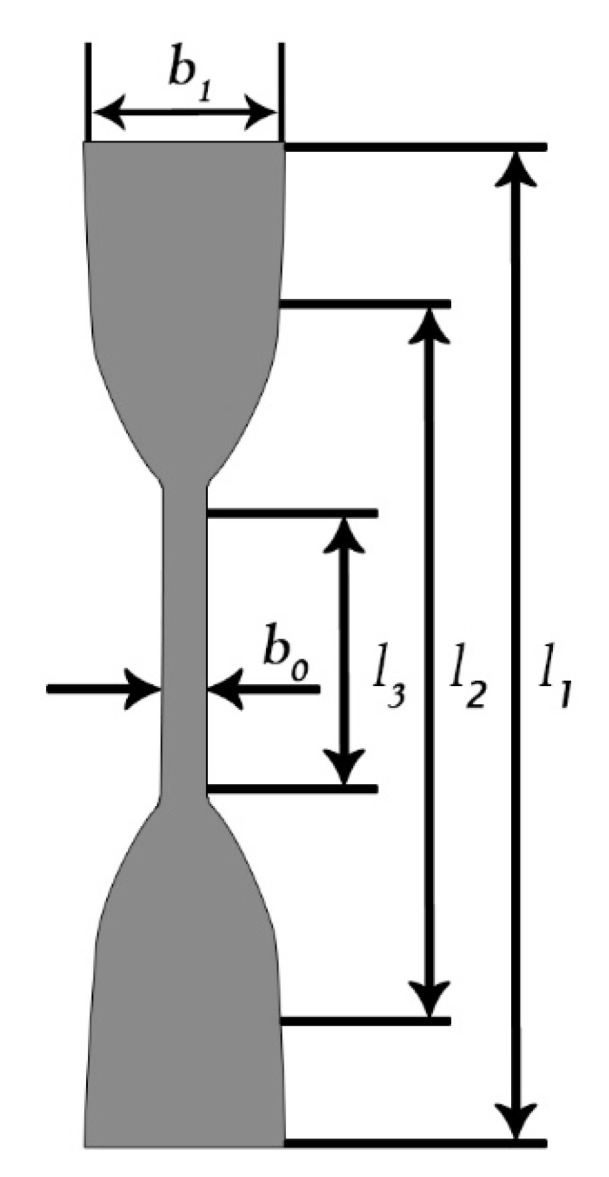
Test sample general view. Type III.

**Figure 4 materials-14-02376-f004:**
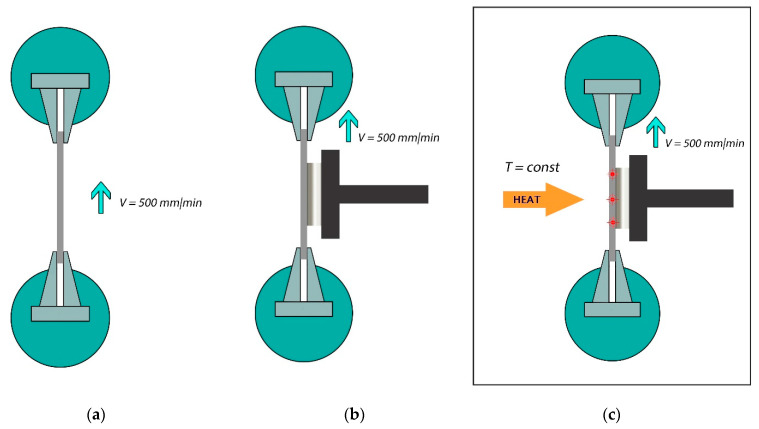
Experimental scheme for evaluating dynamic and mechanical properties of the MAE: (**a**) study without external influence; (**b**) study under the influence of an external magnetic field; (**c**) study under the influence of a combination of external magnetic and temperature fields.

**Figure 5 materials-14-02376-f005:**
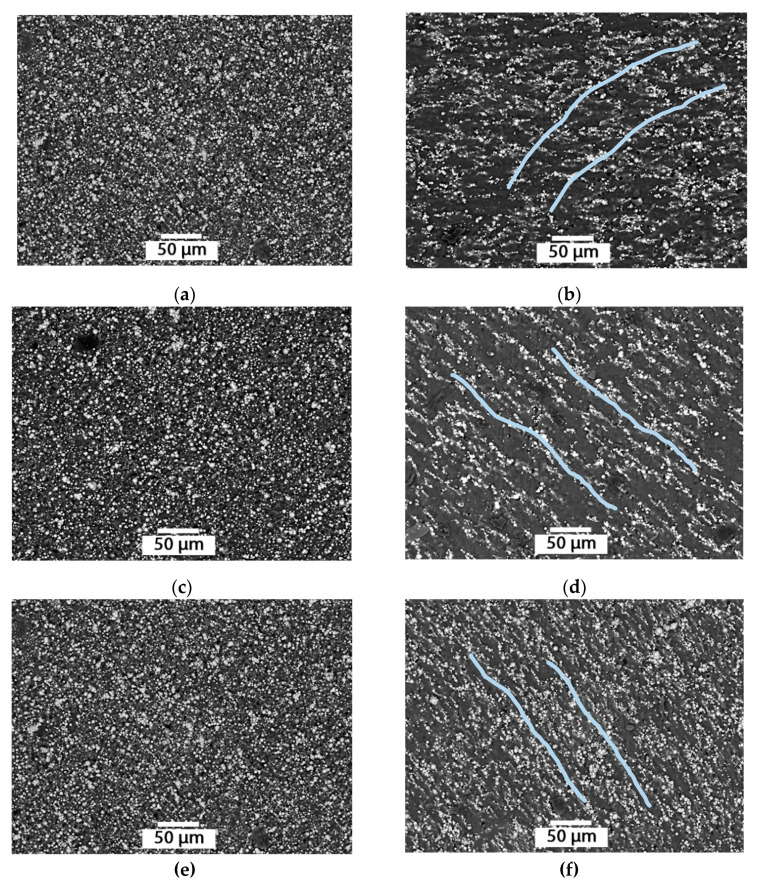
Internal structure of MAE samples: (**a**,**c**,**e**) isotropic structure; (**b**,**d**,**f**) anisotropic structure; 50%, 60%, 70% filling agent content.

**Figure 6 materials-14-02376-f006:**
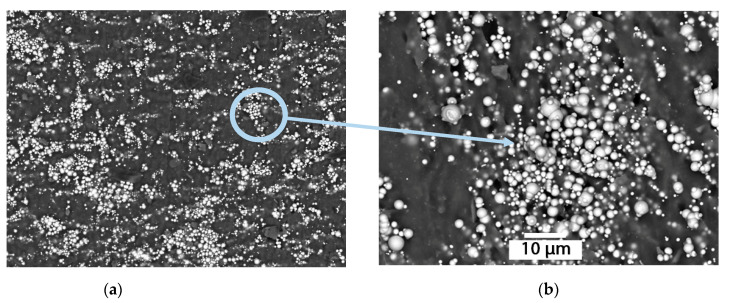
Agglomeration areas formation in course of MAE polymerization with R-10 filler in an external magnetic field, 70%: (**a**) increase up to 50 μm; (**b**) increase up to 10 μm.

**Figure 7 materials-14-02376-f007:**
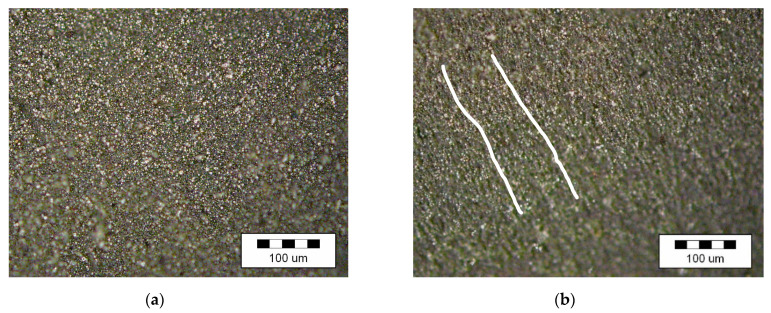
Internal structure of MAE samples with a filling agent content of 70%: (**a**) isotropic structure; (**b**) anisotropic structure.

**Figure 8 materials-14-02376-f008:**
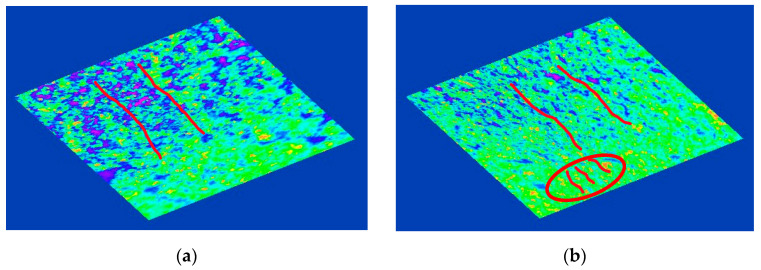
Agglomeration areas formation in course of MAE polymerization with atomized powdered iron filler in an external magnetic field: (**a**) 50%; (**b**) 70%.

**Figure 9 materials-14-02376-f009:**
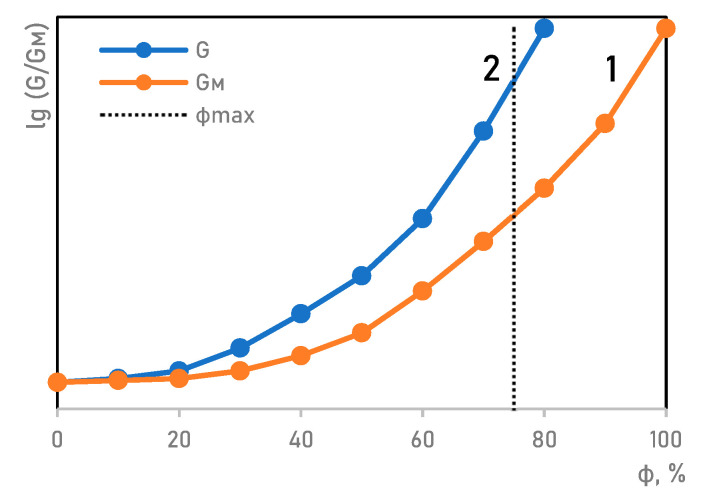
Dependence between the relative elastic modulus G/GM and the filling agent content φ: (**1**) design curve disregarding the limiting particle packing; (**2**) curve including the limiting (dense) particle packing φ_max_.

**Figure 10 materials-14-02376-f010:**
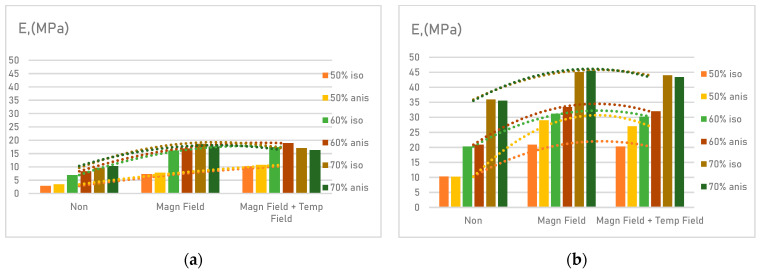
Assessment of MAE characteristics—elastic modulus: (**a**) samples with R-10 filler; (**b**) samples with atomized powdered iron filler.

**Figure 11 materials-14-02376-f011:**
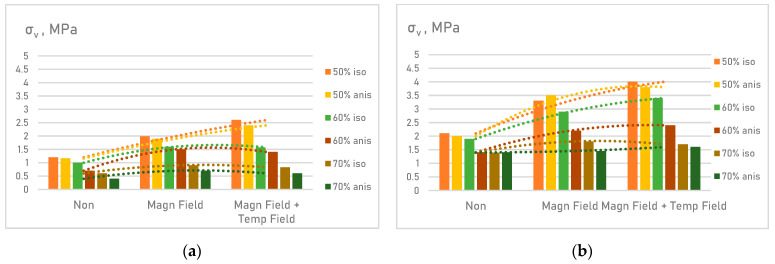
Research on the MAE dynamic and mechanical properties—tensile strength: (**a**) samples with P10 filling agent; (**b**) samples with atomized powdered iron filling agent.

**Table 1 materials-14-02376-t001:** Main magnetic properties of NdFeB magnetic bar.

Parameter	Value
Residual magnetic induction, Br	1.21 T.
Coercive force by magnetization, Hcb	>876 kA/m
Coercive force by induction, Hcj	>955 kA/m
Maximum magnetic energy, BH	263–287 kJ/m^3^
Range of operating temperatures	−60–+80 °C

**Table 2 materials-14-02376-t002:** Dimensions of the samples for research.

Parameter	Value, mm
Overall length, l_1_	75.0
Width of the wide part, b_1_	12.5
Length of the narrow part, l_3_	25.5
Width of the narrow part, b_0_	4.0
Distance defining a radius of curvature, l_2_	50.0
Distance between marks, l_0_	20.0
Simple thickness	2.0

## Data Availability

The data that support the findings of this study are available on request from the corresponding author. The data are not publicly available due to privacy or ethical restrictions.

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
