# Peer review of "Research on Dynamic and Mechanical Properties of Magnetoactive Elastomers with High Permeability Magnetic Filling Agent at Complex Magneto-Temperature Exposure"

_materials, 2021, doi:10.3390/ma14092376_

Round 1

Reviewer 1 Report

Scientific approach and procedure:

The paper reports experimental investigations of the strength as well as the dynamic properties of magneto-rheological (active) elastomers based on highly permeable magnetic filling agents. Additionally, the magneto-rheological effect (stiffening within a magnetic field) is examined. An emphasis is given to the understanding of these type of elastomeric composites’ micro-structures and their effect on the macroscopic mechanical behavior. Therefore, the particles were distributed randomly (isotropic mechanical behavior) and oriented by an external magnetic field (anisotropic behavior).

Remarks:

I appreciate the changes, but some fundamental descriptions would be very useful. For example, the loss modulus versus storage modulus diagram (as shown in Tschoegl, Knauss &Emri, Mech Time-Depend Mater 6(1):53-99 or in Çakmak, Fischlschweiger, M., Graz, I., & Major, Z. (2020). Polymers, 12(11), 2440, MDPI.; ) would be very helpful to understand the loading rate - temperature dependency of the material (within as well as w/o the magnetic field). If you have not such data, please address this interaction of viscoelasticity and magnetorheological effect in the introduction. I have had provided some literature therefore. 

  • Figure 8: include a scale

https://doi.org/10.14311/APP.2016.3.0007

(Çakmak, Fischlschweiger, M., Graz, I., & Major, Z. (2020). Polymers, 12(11), 2440, MDPI.; Tschoegl, Knauss &Emri, Mech Time-Depend Mater 6(1):53-99)

My overall suggestion is that the content of the paper is relevant to the scope and the audience of the journal, but has to be minor revised (specially in terms of viscoelasticity of magnetoactive elastomers).

Author Response

We thank the reviewer for the feedback on the work.

We answer according to the sequence of comments:

  1. "I appreciate the changes, but some fundamental descriptions would be very useful. For example, the loss modulus versus storage modulus diagram (as shown in Tschoegl, Knauss &Emri, Mech Time-Depend Mater 6(1):53-99 or in Çakmak, Fischlschweiger, M., Graz, I., & Major, Z. (2020). Polymers, 12(11), 2440, MDPI.; ) would be very helpful to understand the loading rate - temperature dependency of the material (within as well as w/o the magnetic field). If you have not such data, please address this interaction of viscoelasticity and magnetorheological effect in the introduction. I have had provided some literature therefore."

Answer: 

The authors did not consider this issue separately. Consideration of the results of relevant studies has been added to the work. “Due to the elastomer’s viscoelastic material behavior, the surface properties depend on loading speed and temperature. Different peeling speeds result in different adherence strength of an interface between the gripper and the substrate. The studies were carried out at the same speed to exclude the influence of this effect." A link to the relevant work has been added to the bibliography.

2. Figure 8: include a scale

Answer: Adding a scale is not possible, since the figure shows a section. The figure is intended to demonstrate the severity of the anisotropy of the sample.

Reviewer 2 Report

see attached document!

Author Response

We thank the reviewer for the feedback on the work.

In line 77, the corresponding changes in terminology were made.

Reviewer 3 Report

Thank you for making further improvements. Still, errors exist in the exponential number format, e.g. row 90. Maybe the editors can fix this, there are not many occurrences of this mistake. Overall, I like the paper and would vote for "accept".

Author Response

We thank the reviewer for the prepared feedback on the work.

An edit was made to the exponential value on line 91.

Reviewer 4 Report

The paper entitled "Research on Dynamic and Mechanical Properties of Magnetoactive Elastomers with High Permeability Magnetic Filling Agent at Complex Magneto-Temperature Exposure" by Maria Vasilyeva, Dmitry Nagornov and Georgiy Orlov considers magnetically active elastomer as a promising material for manufacturing a working channel of a magnetic pump unit. In the study, it was shown that as the combined effect of magnetic and temperature fields, the elasticity and strength of materials increases by an average of 30% in comparison with the material without external influence. Based on the results obtained, the composition and structural organization of the final material was substantiated, which has the best set of 18 parameters. The paper can be interesting for readers of Materials.

I recommend this paper for publication, it is well written and can be published as-is.

Author Response

We thank the reviewer for the positive feedback on the work.

Reviewer 5 Report

In this study, the mechanical and dynamic mechanical properties of magnetoactive elastomers based on soft magnetic fillers were investigated. The results indicate that using a filler with a relatively coarse fraction, the material has more stable dynamic and mechanical characteristics. This research work and the results may be interested by broad range of readers. After careful consideration, the manuscript is recommended for publishing in the journal Materials after addressing following comments.

  1. In the Conclusion part, the authors claimed that “the method of polymerization of an elastomer is suitable for use as a material for the working channel of a magnetic pump unit”. What is the method of polymerization of an elastomer and how it influences the dynamic mechanical properties of the resulted magnetic elastomer?
  2. In addition to the particle size of fillers, the interfacial interaction between filler and matrix is also crucial to the properties of the composite. It is better to pay more attention to the interfacial interaction between fillers and matrix.
  3. How to prepare the samples with isotropic and anisotropic structures? The preparation procedure should be depicted in detail.   
  4. There are so many morphological observations in the manuscript, please remove some of it to Supporting Information.

Author Response

  1. In the Conclusion part, the authors claimed that “the method of polymerization of an elastomer is suitable for use as a material for the working channel of a magnetic pump unit”. What is the method of polymerization of an elastomer and how it influences the dynamic mechanical properties of the resulted magnetic elastomer?

Answer: The goal of the authors was to determine what composition and what structure the final material should have in order to meet the required elastic-strength parameters. The structure of the material will depend on the polymerization procedure (whether a magnetic field is applied to the solidified mixture or not). According to the results of the study, the best relative results were shown by samples polymerized without the application of an external magnetic field and having, accordingly, an isotropic structure.

  1. In addition to the particle size of fillers, the interfacial interaction between filler and matrix is also crucial to the properties of the composite. It is better to pay more attention to the interfacial interaction between fillers and matrix.

Answer: The authors were interested in the question of justifying the choice from the existing variety of fillers of the sample that would provide the best elastic and strength properties of the final material, provided that the level of its magnetic susceptibility is maintained, as well as the very possibility of manufacturing a composite elastomer based on it.

  1. How to prepare the samples with isotropic and anisotropic structures? The preparation procedure should be depicted in detail.  

Answer: This procedure is not new and is presented in a significant number of earlier studies (references to them are present in the list of references). A feature of the manufacture of an isotropic elastomer in this work is the use of a permanent magnet as a source of a magnetic field. The procedure is described in the appropriate section.

  1. There are so many morphological observations in the manuscript, please remove some of it to Supporting Information.

Answer: If the reviewer does not consider the implementation of this recommendation as a fundamental condition for the publication of the work, the authors would like to leave the sequence of presentation of the material in the article unchanged.

Reviewer 6 Report

This manuscript contains good results and can be considered for publication after major revision:
1. Abstract should highlight the main message of the work and quantitative analysis of results. Revise. 
2. Scales in Figures 10, 11, ... should be revisited. The present form is not suitable. 
3. Correlation between DMA and agglomeration should be reinforced in discussion. 
4. Contribution of  Vasilyeva M., as stated in the manuscript, is highly distinctive. It appears that others did nothing comparatively. Is it logical? 

Author Response

We thank the reviewer for the feedback on the work.

Abstract should highlight the main message of the work and quantitative analysis of results. Revise.

Answer: The presented Abstract section contains information about the main idea of the work, presents the main circumstances of the study and indicates what conclusions readers can get acquainted with after reading the manuscript. The authors would like to bring the obtained quantitative values into the Conclusions section.

2. Scales in Figures 10, 11, ... should be revisited. The present form is not suitable. 

Answer: Unfortunately, the reviewer did not indicate what exactly he considered unacceptable in the scales in Figures 10 and 11. These figures have a comparative principle of construction: the ordinate represents the value of the parameter under study, and the abscissa shows a list of samples from which these values were obtained.
Taking into account the above explanation, we have left the figures unchanged.

3. Correlation between DMA and agglomeration should be reinforced in discussion. 

Answer: The authors did not make an in-depth analysis of the mechanism of the formation of foci of agglomeration, just as they did not study separately the changes in the properties of the material in dynamics during the formation of agglomerates. Conclusions regarding their influence on the elastic-strength properties of the material are made on the basis of the known properties of composite polymers and the effect on them of the concentration and distribution of the filler material in the matrix.

4. Contribution of  Vasilyeva M., as stated in the manuscript, is highly distinctive. It appears that others did nothing comparatively. Is it logical? 

Answer: Did it - pointed it out. Everything is logical.

Round 2

Reviewer 6 Report

Authors MUST appreciate and accept the comments of reviewers! The introduction is very short and the story is incomplete. what's new in this study??

Author Response

Dear Reviewer!

We carefully read the content of your review and answered the following questions in detail:

1. The annotation section is supplemented with the presentation of the results.

2. Each column represents the quantitative value of the corresponding parameter of each sample. Columns are grouped depending on the conditions of the study: in the absence of additional exposure (Non), under the influence of an external magnetic field (Magn Field), with the combined effect of an external magnetic field and temperature exposure (Ðœagn Field + Temp Field). This makes it possible to compare changes in parameters in dynamics.
We could not choose a more visual way of presenting the material.

3. The authors did not make an in-depth analysis of the mechanism of the formation of foci of agglomeration. And they did not study separately the changes in the properties of the material in dynamics during the formation of agglomerates. Conclusions regarding their influence on the elastic-strength properties of the material are made based on the known properties of composite polymers and the effect on them of the concentration and distribution of the filler material in the matrix. Separate conclusions regarding the influence of sintering on the mechanical characteristics of the material were not made. The material was assessed entirely from the point of view of providing the necessary strength properties for practical use in a particular object. 
This will be considered by the authors as an idea for an additional deeper study of the elastomer material.

We hope our reasoning will convince you.

We respect each reviewer's work and deeply regret if a reviewer gets the impression that their work has been underestimated. This is our fault.
If at this stage we have not been able to fully unleash the potential of the recommendations, we will work to ensure that they find their application in the future.

Round 3

Reviewer 6 Report

NtR